# Prevalence and sociodemographic determinants of suboptimal glycemic control in persons with diabetes in Ghana: A systematic review and meta-analysis

Emmanuel Ekpor[1], Dorothy Wilson[2], Eric Peprah Osei[3], Bernard Abeiku Mensah[4], Samuel Akyirem[4]*

1 School of Psychology, Deakin University, Geelong, Victoria, Australia, 2 School of Nursing and Midwifery, Kwame Nkrumah University of Science and Technology, Kumasi, Ghana, 3 College of Nursing, University of Illinois Chicago, Chicago, Illinois, United States of America, 4 Yale School of Nursing, Yale University, New Haven, Connecticut, United States of America

* samuel.akyirem@yale.edu

## Abstract

Effective glycemic control is a cornerstone of diabetes management, essential for reducing the risk of complications and mortality. However, in Ghana, persistent limitations in diabetes management capacity present significant challenges to meeting recommended glycemic targets. This systematic review synthesizes the available evidence on the prevalence and sociodemographic determinants of suboptimal glycemic control among individuals with diabetes in Ghana. Relevant observational studies were obtained through a systematic search conducted on PubMed, Medline, Embase, Global health, Scopus, and Web of Science, from their inception to November 29, 2024. A random-effects meta-analysis was used to estimate the pooled prevalence of suboptimal glycemic control, accounting for heterogeneity across studies. Subgroup analyses were performed to explore potential sources of variability. We assessed publication bias statistically using Egger's regression and Begg's rank correlation test. Out of 1390 articles screened, 28 meet the inclusion criteria, comprising a total of 11,242 participants. The pooled prevalence of suboptimal glycemic control was 67.6% (95% CI: 64.2–70.8). When stratified by glycemic control measures, the prevalence was 69.2% (95% CI: 62.5–75.2) for fasting blood glucose levels ≥7.0 mmol/L and 66.9% (95% CI: 62.5–70.9) for hemoglobin A1c levels ≥7.0%. Sociodemographic factors such as age, income, gender, and ethnicity were found to be associated with suboptimal glycemic control. These findings underscore the substantial burden of suboptimal glycemic control among individuals with diabetes in Ghana, with over two-thirds not meeting recommended targets. There is an urgent need for targeted, context-specific interventions to address both clinical and systemic barriers to effective diabetes management in Ghana.

**Data availability statement:** All relevant data are within the paper and its Supporting Information files. All articles used for this study can be accessed online by following the references.

**Funding:** The author(s) received no specific funding for this work.

**Competing interests:** The authors have declared that no competing interests exist.

## Introduction

Diabetes mellitus is a chronic metabolic disorder marked by persistently high blood glucose levels due to impairments in insulin production, function, or both. The global prevalence of diabetes has escalated dramatically, with an estimated 537 million adults affected in 2021 [1]. Current research projections reveal this crisis will intensify, particularly in Africa, where there is an expected 134% increase in diabetes cases by 2045 [1]. Concerningly, this growing burden is accompanied by a rise in diabetes-related complications, such as cardiovascular diseases, which significantly contribute to morbidity and mortality across the continent [2,3].

Achieving effective glycemic control is fundamental in diabetes management, as it plays a critical role in minimizing complications, improving health outcomes, and enhancing the quality of life for individuals living with the condition [4]. However, a global trend reveals that fewer than 50% of individuals with diabetes achieve recommended glycemic targets [5]. In the African context, systemic barriers (including limited access to quality diabetes care) worsen this issue, resulting in persistently high rates of suboptimal glycemic control. For instance, a systematic review from Ethiopia reported that approximately 61% of persons with diabetes did not achieve optimal glycemic control [6]. Similarly, rates of 75.2% and 83.3% were recorded in Senegal [7] and Nigeria [8], respectively. A meta-analysis encompassing 16 sub-Saharan African countries revealed that about 70% of individuals did not achieve glycemic control targets [9].

In Ghana, the diabetes burden mirrors these regional trends. According to the Global Burden of Disease Study, the age-standardized prevalence of diabetes in Ghana was 5.3% in 2021, with projections estimating an increase to 9.5% by 2050 [10]. However, despite this growing prevalence, diabetes management in Ghana is fraught with several challenges. Research highlights considerable diagnostic gaps, with many diabetes cases remaining undetected, which delays timely interventions and increases the risk of complications [11]. Additionally, audits of healthcare facilities reveal a shortage of trained diabetes specialists, coupled with limited availability of essential resources required for effective diabetes management [12]. On a personal level, individuals with diabetes in Ghana often face difficulties in meeting their glycemic needs, due in part to limited access to medications, and the availability of healthy, affordable, and diabetes-friendly food options [13,14].

Given the growing burden of diabetes in Ghana, it is essential to understand the health state of affected individuals. Glycemic control, when not achieved, poses significant risk for worse health outcomes. However, despite its importance, there is no consolidated evidence on the prevalence of suboptimal glycemic control among persons with diabetes in Ghana. By focusing on Ghana, we can conduct granular analysis to highlight within-country regional differences in glycemic control. Moreover, while research from other regions has identified significant associations between glycemic control and sociodemographic factors such as gender, education, and residence [6], evidence specific to Ghana remains scarce. Understanding these factors is vital for developing policies and interventions that cater to the unique needs of individuals living with diabetes. This study, therefore, aimed to estimate the

prevalence and examine the sociodemographic determinants of suboptimal glycemic control among individuals with diabetes in Ghana. By synthesizing available evidence, this study seeks to provide actionable insights that will guide healthcare policies, enhance diabetes management strategies, and ultimately improve health outcomes for individuals living with diabetes in Ghana.

## Materials and methods

This systematic review adhered to the Joanna Briggs Institute (JBI) methodological guidelines for observational epidemiological studies on prevalence and cumulative incidence data [15] and was reported in accordance with the Preferred Reporting Items for Systematic Reviews and Meta-Analyses (PRISMA) guidelines [16] (S1 Checklist). The protocol for this review was prospectively registered with PROSPERO (CRD42024622708).

### Search strategy

A systematic search for relevant studies was conducted on PubMed, Medline, Embase, Global health, Scopus, and Web of Science, from their inception to November 29, 2024. Other sources were also explored to ensure the exhaustiveness of our search. This included Africa-specific research repositories (e.g., African Journals Online), reference list of included articles, and forward citation tracking on google scholar. As shown in S1 Text, the search strategy utilized a combination of keywords and indexed terms in relation to "diabetes", "glycemic control", and "Ghana". The Boolean operators ("AND," "OR") were used to form a search string for the various databases. No language restrictions were applied to the search.

### Inclusion and exclusion criteria

Studies were included if they met the following criteria: (1) focused on adults with diabetes in Ghana, (2) reported the prevalence of "poor" or suboptimal glycemic control, and (3) were observational in design (cross-sectional, case-control, or cohort studies). Studies that did not explicitly define the threshold for suboptimal glycemic control, but stratified glycemic levels into categories that allowed identification of suboptimal control based on recommended guidelines, were included. Studies were excluded if they were access restricted, reviews, and articles in the grey literature (e.g., preprints, thesis, conference proceeding).

### Study selection and quality assessment

Based on the predefined eligibility criteria, screening for relevant studies was performed on Rayyan. The screening was performed in two stages; first by reviewing titles and abstracts, followed by a detailed evaluation of full-text articles. Relevant information from the included studies were extracted using a data extraction matrix in Microsoft Excel. Extracted data included study characteristics (author, year, location, study design, sample size), participant characteristics (sociodemographic variables, diabetes type, diabetes duration), and data on glycemia, including methods of assessment and prevalence of suboptimal glycemic control. The screening and data extraction was performed in duplicates by two reviewers independently, with discrepancies resolved through discussion with a third reviewer.

Quality assessment of each included study was performed using the JBI checklist for prevalence studies [15]. This checklist consists of nine key questions designed to evaluate the methodological limitations of studies, with response options including "Yes", "No", "Unclear", and "Not applicable" (S1 Table). Based on the number of "Yes" responses, studies were categorized as having a low risk of bias (7–9 "Yes" responses), moderate risk of bias (5–6 "Yes" responses), or high risk of bias (<5 "Yes" responses).

### Data analysis

The meta-analysis was conducted using the meta package in R. Pooled prevalence estimate of suboptimal glycemic control was calculated using a DerSimonian-Laird random-effects model, which accounts for variability both within and

between studies. The prevalence estimates were presented with corresponding 95% confidence intervals (CIs) to evaluate the precision of the results. Heterogeneity across studies was assessed using the I² statistic, with scores of 25%, 50%, and 75% representing low, moderate, and high heterogeneity, respectively [17]. Subgroup analyses were conducted to explore potential sources of heterogeneity by stratifying studies based on geographical belts in Ghana (Northern, Middle, and Coastal), gender, study design, diabetes type, and glycemic control metrics. To evaluate the robustness of the pooled prevalence estimates, a leave-one-out sensitivity analysis was conducted. Publication bias was evaluated both visually, using a funnel plot, and statistically, using Egger's regression test [18] and Begg's rank correlation test [19]. If significant publication bias was detected (p < 0.05), the pooled estimate was adjusted using a nonparametric trim-and-fill analysis to account for potentially missing studies [20]. As a secondary objective, a narrative synthesis was performed to summarize the findings of studies reporting adjusted effect measures of sociodemographic factors associated with suboptimal glycemic control.

## Results

The systematic search retrieved 1390 references, consisting of 1376 records from electronic databases and 14 from other sources. After removing duplicates, 635 articles were screened by the title and abstract. Of these, 588 were excluded, with the majority (58%) not addressing the outcome of interest. Full-text screening was conducted on 47 articles, resulting in the exclusion of 19 due to duplicate publications, irrelevant outcomes, irrelevant study designs, and grey literature. Ultimately, 28 studies were included in the systematic review (Fig 1).

### Characteristics of included studies

Table 1 presents details of studies included in this review. The majority of articles included in this review employed cross-sectional design (n = 25), with the remaining being cohort and case control. Articles were published between 2013–2024, majority (n = 16) of which were published in the recent five years. The studies panned various regions, with Ashanti being the most represented (n = 11) [21–31], followed by Greater Accra (n = 5) [32–36], Volta (n = 5) [37–41], Northern (n = 2) [42,43], and one each from Bono [44], Central [45], Eastern [46], and Upper East [47]. One study was multi center, involving participants recruited from diverse regions [48]. This review included a total of 11242 participants, with the sample size ranging from 115 to 2593. Twenty-seven studies reported the gender proportion of participants, with majority being female (66.1%). Participants were on average 57 years (n = 22), and had lived with diabetes between 7 and 10 years. Glycemic control was assessed using either hemoglobin A1c (HbA1c) (n = 13) or fasting blood glucose (FBG) (n = 14), with one study using both metrics [24]. Various cut-off for suboptimal glycemic control was used, with majority of studies using >7.0/ ≥ 7.0 mmol/L (n = 14) as threshold for FBG, while> 7.0/ ≥ 7.0% (n = 10) was used for HbA1c.

### Quality assessment of included studies

The quality scores of the included studies ranged from 4 to 9, with an average score of 6.9. All studies were assessed as having a low to moderate risk of bias, except for one study, which demonstrated a higher risk with a quality score of 4. The primary methodological shortcomings across the studies pertained to several areas. These included the appropriateness of participant sampling, adequacy of the sample size, lack of clarity regarding whether the condition was measured in a standard and reliable manner for all participants, and insufficient detail about participant response rates (S1 Table).

### Prevalence of suboptimal glycemic control

The prevalence of suboptimal glycemic control was pooled across studies regardless of the specific metric used to assess glycemic control. For the study by Adua et al. [24], which reported both FBG and HbA1c, the average prevalence from the two metrics was included in the overall analysis to prevent double-counting participants. Variations between the metrics were accounted for separately in the subgroup analysis.

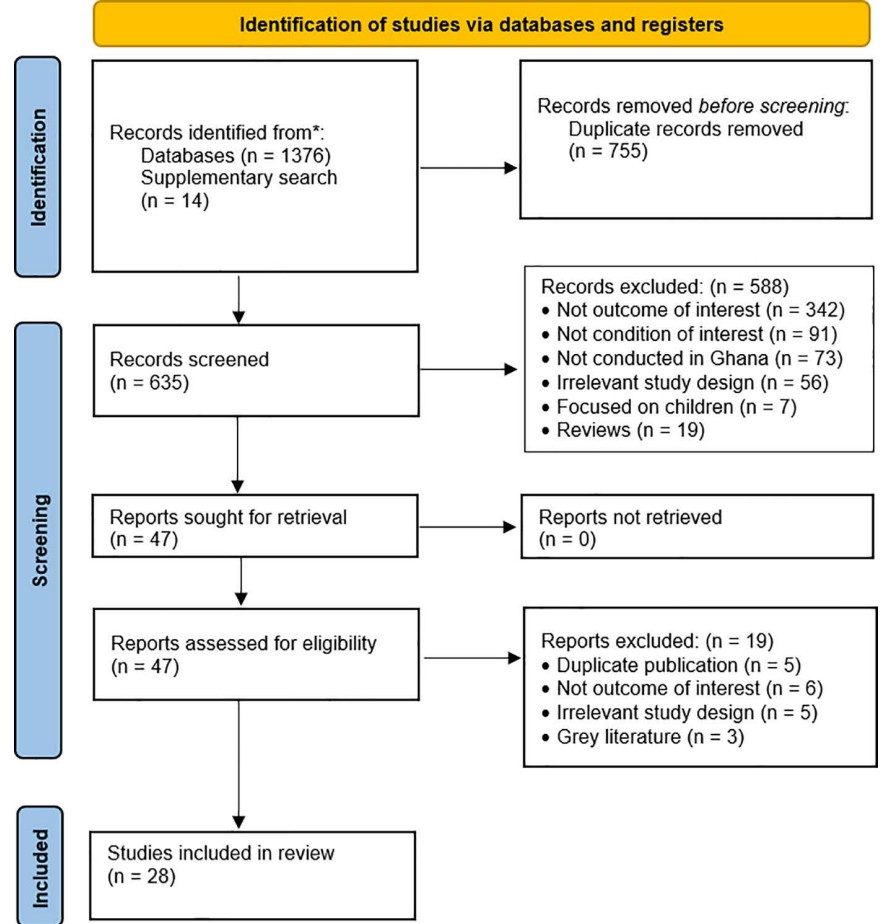

**Fig 1. PRISMA flow chart summarizing the article selection process.**

The pooled prevalence of suboptimal glycemic control was 67.6% (95% CI: 64.2–70.8). Notably, there was substantial heterogeneity among the included studies ($I^2 = 92\%$, $p < 0.01$) (Fig 2). To ensure comparability of our findings, the prevalence estimate was pooled based on the most commonly used cut-off for suboptimal glycemic control. The result revealed a 69.2% (95% CI: 62.5–75.2) prevalence for studies using >7.0/≥7.0 mmol/L as threshold for FBG, whiles those using >7.0/≥7.0% as threshold for HbA1c recorded 66.9% (95% CI: 62.5–70.9).

As shown in Table 2 and S1 Fig, the subgroup analysis revealed notable variations in the prevalence of suboptimal glycemic control; however, only the test for subgroup differences based on diabetes type was statistically significant ($p = 0.04$). Specifically, the highest prevalence was found among individuals with unspecified diabetes type (either type 1 or type 2), at 79.1% (95% CI: 66.8–87.6), compared to 65.6% (95% CI: 61.7–69.2) among those diagnosed with type 2 diabetes. Regional analysis highlighted that the Coastal Belt of Ghana exhibited the highest prevalence of suboptimal glycemic control at 70.4% (95% CI: 62.8–77.1), while the Middle Belt had the lowest prevalence of 65.6% (95% CI: 61.3–69.6). Gender differences showed only marginal variation in prevalence estimates, with males exhibiting a prevalence of 65.5% (95% CI: 58.8–71.6) and females 64.5% (95% CI: 57.2–71.2). When stratified by study design, cohort studies recorded the highest prevalence at 72.5% (95% CI: 57.2–83.8), followed by cross-sectional studies at 67.4%, and case-control studies at 61.0%. Regarding the metrics used to assess glycemic control, FBG yielded a prevalence of 69.7% (95% CI: 63.3–75.4), while HbA1c showed a prevalence of 65.4% (95% CI: 61.8–68.9).

**Table 1. Characteristics of included studies.**

| Author (year) | Diabetes type | Design | Region | Sample size | Female % | Mean age | Mean diabetes duration | Metric for PGC | FBG Cut-off (mmol/L) | HbA1c Cut-off (%) |
|---|---|---|---|---|---|---|---|---|---|---|
| Afaya (2020) | T2D | Cross sectional | Northern | 330 | 68.2 | 57.5 | NR | FBG | ≥7.0 | NR |
| Sefah (2020) | T2D | Cross sectional | Volta | 400 | 72.0 | NR | NR | FBG | ≥7.0 | NR |
| Botchway (2022) | T2D | Cross sectional | Ashanti | 254 | 59.5 | 62.90 | 13.14 | HbA1C | NR | > 7.0 |
| Mobula (2018) | T2D | Cross sectional | Multi | 1226 | 77.5 | 57.0 | 9.4 | HbA1C | NR | ≥7.0 |
| Alor (2023) | T2D | Cross sectional | Volta | 310 | 65.8 | 57.8 | 7.4 | FBG | >7.2 or < 3.9 | NR |
| Agyekum (2023) | T2D | Case control | Bono | 200 | 62.0 | 52.0 | NR | HbA1C | NR | > 7.0 |
| Fiagbe (2017) | Unspecified | Cross sectional | Volta | 220 | 75.9 | 60.59 | NR | FBG | ≥7.0 | NR |
| Antwi-Baffour (2023) | T2D | Cross sectional | Greater Accra | 384 | 67.2 | 59.51 | NR | HbA1C | NR | ≥7.0 |
| Sarfo (2020) | T2D | Cross sectional | Ashanti | 279 | 53.4 | 61.4 | NR | FBG | ≥7.0 | NR |
| Alhassan (2022) | T2D | Cross sectional | Greater Accra | 329 | 56.2 | 57.5 | NR | FBG | ≥7.0 | NR |
| Agyemang-Yeboah (2019) | T2D | Cross sectional | Ashanti | 405 | 80.0 | 58.5 | 6.4 | HbA1C | NR | >7 |
| Adua (2017) | T2D | Cohort | Ashanti | 241 | 59.2 | 57.82 | NR | FBG, HbA1c | >7 | >7.2 |
| Osei-Yeboah (2019) | T2D | Cross sectional | Volta | 150 | 72.7 | NR | NR | FBG | >7 | NR |
| Asamoah-Boakye (2017) | T2D | Cross sectional | Ashanti | 152 | 75.7 | 55.5 | NR | FBG | ≥7.0 | NR |
| Brenyah (2013) | T2D | Cross sectional | Ashanti | 341 | NR | 54.9 | 5.8 | FBG | ≥7.0 | NR |
| Adu (2019) | T2D | Cross sectional | Eastern | 324 | 75.9 | 57.0 | NR | FBG | ≥7.0 | NR |
| Sarfo-Kantanka (2018) | T2D | Cross sectional | Ashanti | 780 | 57.7 | 57.4 | 9.8 | HbA1C | NR | > 7.0 |
| Lokpo (2022) | T2D | Cross sectional | Volta | 210 | 54.3 | 49.98 | NR | FBG | >7 | NR |
| Adjei (2024) | T2D | Cross sectional | Greater Accra | 227 | 78.0 | 60.76 | NR | HbA1C | NR | ≥7.0 |
| Adu (2024) | T2D | Cross sectional | Ashanti | 400 | 56.5 | NR | NR | HbA1C | NR | ≥7.0 |
| Mogre (2014) | T2D | Cross sectional | Northern | 300 | 77.0 | 56.21 | 5.23 | FBG | ≥7.0 | NR |
| Swaray (2023) | Unspecified | Cross sectional | Ashanti | 2593 | 64.0 | 54.8 | NR | HbA1C | NR | > 7.0 |
| Adong (2024) | Unspecified | Cohort | Central | 251 | 72.9 | NR | NR | FBG | ≥7.0 | NR |
| Sisu (2021) | Unspecified | Cross sectional | Upper East | 152 | 47.4 | NR | NR | FBG | ≥7.0 | NR |
| Yorke (2024) | T2D | Cross sectional | Greater Accra | 156 | 76.3 | NR | NR | HbA1C | NR | > 7.0 |
| Djonor (2021) | T2D | Cross sectional | Greater Accra | 271 | 71.6 | 56.6 | NR | HbA1C | NR | ≥8.0 |
| Addai-Mensah (2019) | T2D | Cross sectional | Ashanti | 242 | 31.4 | 58.74 | 4.87 | HbA1C | NR | ≥ 8.0 |
| Apini (2018) | T2D | Cross sectional | Ashanti | 115 | 71.3 | 58.4 | 6.7 | HbA1C | NR | ≥ 6.5 |

Abbreviations: FBG, fasting blood glucose; HbA1c, hemoglobin A1C; NR, not reported; T2D, type 2 diabetes.

## Sensitivity analysis

A leave-one-out sensitivity analysis was performed to systematically remove one study at a time to ensure that the overall results were not disproportionately influenced by any single study. The result revealed a small variation in the pooled prevalence estimate, which ranged from 66.8% to 68.4% across the 28 iterations. As shown in Fig 3, the confidence intervals for the pooled prevalence remained largely overlapping throughout, indicating that the results were not significantly influenced by the exclusion of any individual study.

## Publication bias

A visual inspection of the funnel plot (Fig 4) revealed asymmetry, suggesting the possibility of publication bias. Statistical evaluation using Egger's regression test showed no significant asymmetry (p = 0.3420). However, Begg's rank correlation test suggested evidence of publication bias (p = 0.0362). To address this further, a trim-and-fill analysis was conducted, which identified and imputed four potentially missing studies to achieve a more balanced distribution of effect sizes. After

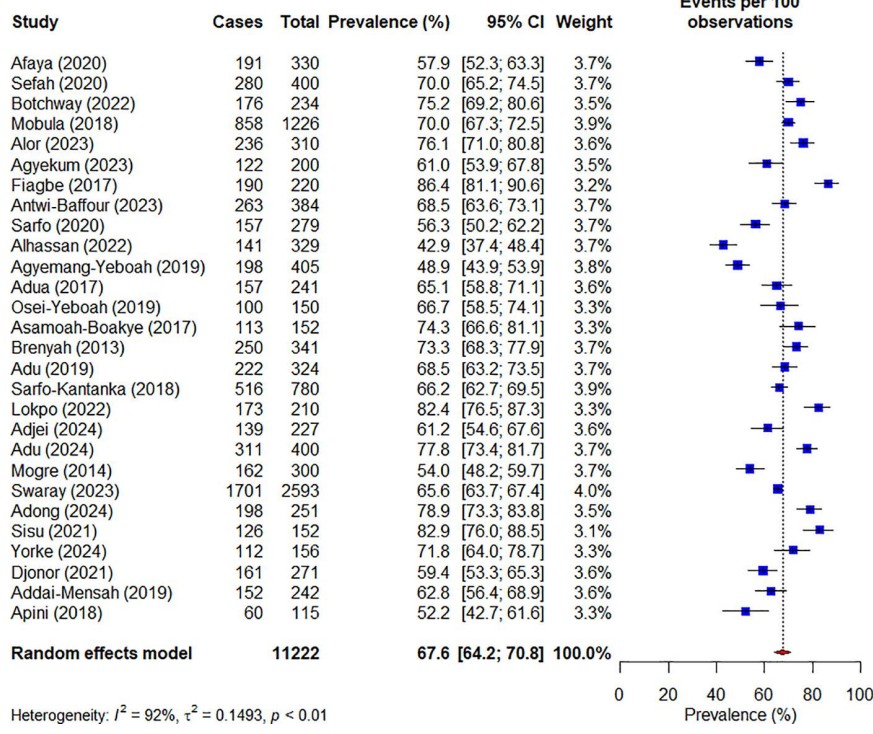

**Fig 2. Forest plot for the prevalence of suboptimal glycemic control.**

**Table 2. Subgroup analysis of prevalence estimates.**

| Subgroup | No. studies | Prevalence (95% CI) | Heterogeneity I² (P value) | Subgroup difference (P value) |
|---|---|---|---|---|
| **Belt** | | | | $X^2 = 3.53$ (0.32) |
| Coastal | 11 | 70.4% (62.8–77.1) | 94% (<0.01) | |
| Northern | 3 | 65.9% (50.1–78.7) | 94% (<0.01) | |
| Middle | 13 | 65.6% (61.3–69.6) | 90% (<0.01) | |
| Mixed | 1 | 70.0% (67.3–72.5) | — | |
| **Study design** | | | | $X^2 = 3.59$ (0.17) |
| Cohort | 2 | 72.5% (57.2–83.8) | 91% (<0.01) | |
| Cross-section | 25 | 67.4% (63.8–70.9) | 93% (<0.01) | |
| Case control | 1 | 61.0% (53.9–67.8) | — | |
| **Metric** | | | | $X^2 = 1.37$ (0.24) |
| FBG | 15 | 69.7% (63.3–75.4) | 94% (<0.01) | |
| HbA1c | 14 | 65.4% (61.8–68.9) | 89% (<0.01) | |
| **Diabetes type** | | | | $X^2 = 4.25$ (0.04) |
| Unspecified | 4 | 79.1% (66.8–87.6) | 95% (<0.01) | |
| Type 2 diabetes | 24 | 65.6% (61.7–69.2) | 92% (<0.01) | |
| **Gender** | | | | $X^2 = 0.04$ (0.84) |
| Male | 11 | 65.5% (58.8–71.6) | 82% (<0.01) | |
| Female | 11 | 64.5% (57.2–71.2) | 93% (<0.01) | |

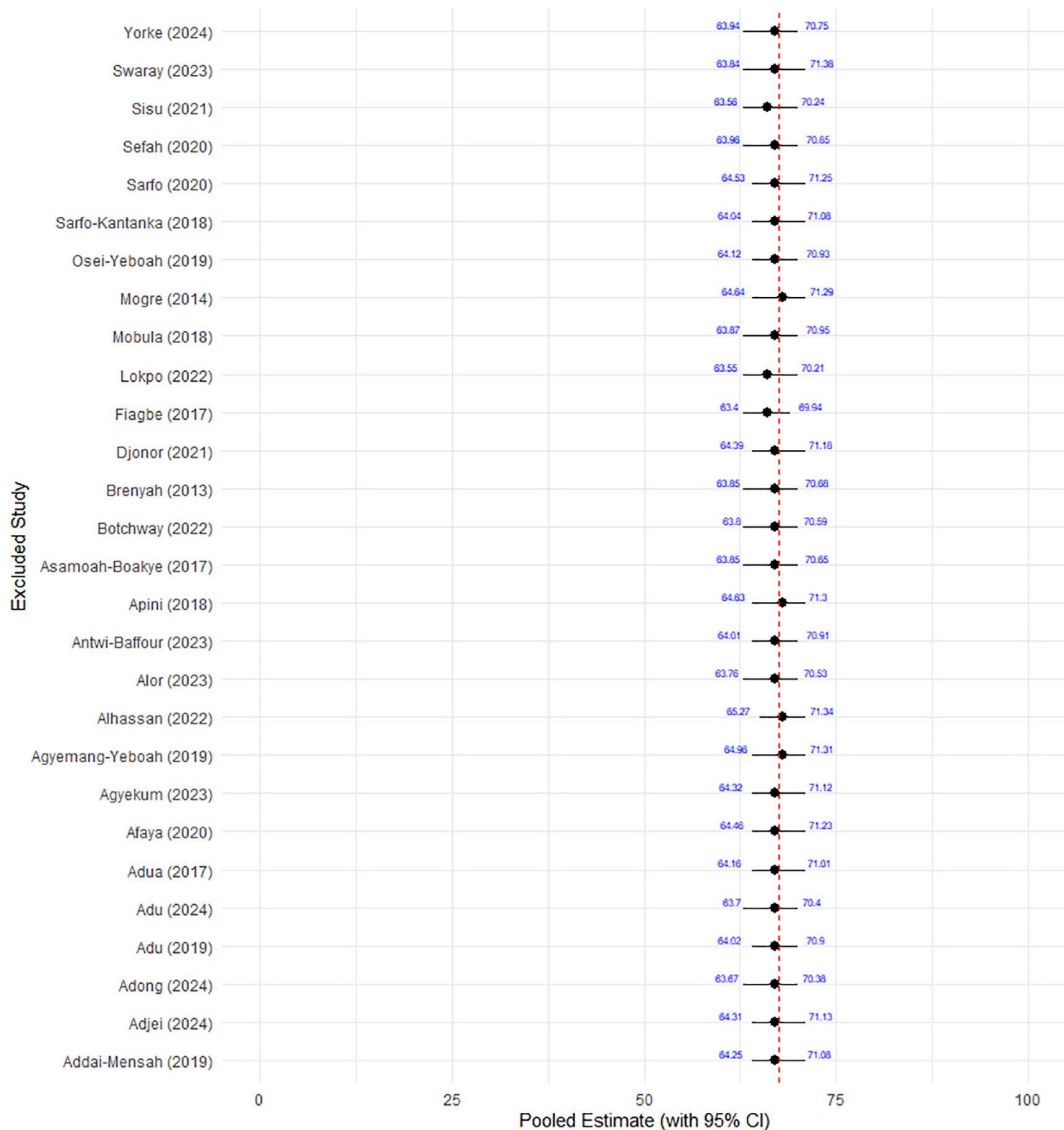

**Fig 3. Plot for leave-one-out sensitivity analysis.**

adjustment, the pooled prevalence of suboptimal glycemic control decreased by 2.8% (S2 Fig), suggesting just a marginal probable overestimation of the pooled prevalence.

## Sociodemographic determinant of suboptimal glycemic control

Few studies (n = 4) reported odds ratios describing the association between sociodemographic characteristics and sub-optimal glycemic control, limiting our ability to perform a meta-analysis. Thus, a narrative synthesis of the finding was performed. It should be noted that, in certain studies, the odds ratios reported were not consistent with the data presented by those studies (e.g., see table 3 of Alor et al. [38] and table 1 of Mobula et al. [48]). In such cases, we recalculated the odds ratio based on the available data. We found that increasing age was significantly associated with lower odds of suboptimal glycemic control [38,48]. Additionally, higher income was associated with greater odds of suboptimal glycemic

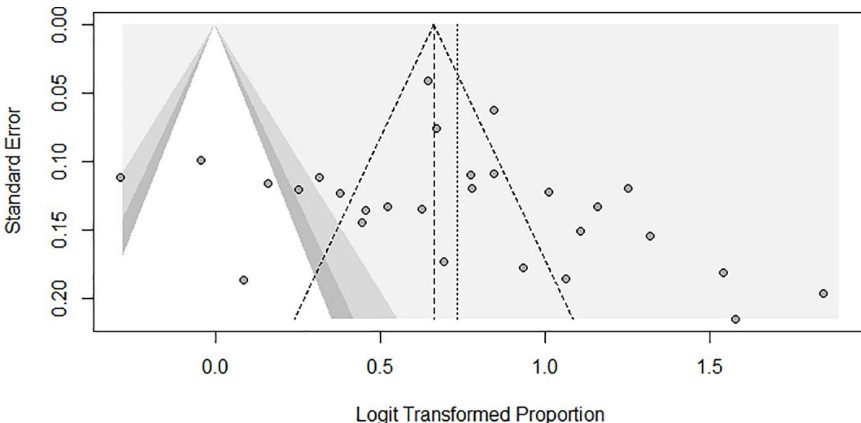

**Fig 4. Funnel plot for publication bias.**

control [33,38]. Women were also less likely to have suboptimal glycemic control [33,48]. Ethnicity was examined in only one study, with the result indicating that Akans had a higher risk of suboptimal glycemic control compared to other ethnic groups [35].

## Discussion

This study aimed to estimate the prevalence and examine the sociodemographic determinants of suboptimal glycemic control among individuals with diabetes in Ghana. By synthesizing data from 28 observational studies, our meta-analysis revealed a high prevalence of suboptimal glycemic control at 67.6%. Geographic variations were observed, with prevalence estimates ranging from 65.6% in the Middle Belt to 70.4% in the Coastal Belt. However, the test for subgroup differences was not statistically significant, indicating that suboptimal glycemic control is a pervasive issue transcending regional boundaries in Ghana.

The high prevalence of suboptimal glycemic control observed in this study aligns with findings from other African countries. For instance, a meta-analysis conducted in Ethiopia reported a prevalence of 61% [6], while a broader study encompassing 16 sub-Saharan African countries found an even higher rate of approximately 70% [9]. Despite the diversity of healthcare systems across the continent, these consistently high rates highlight shared regional challenges in managing diabetes. A nationally representative study spanning 55 countries supports this assertion, revealing that sub-Saharan Africa had the second-lowest coverage of glucose-lowering medications, with only 46.1% of individuals receiving appropriate treatment [49]. Moreover, recent findings from a latent class analysis categorized many African countries as having "limited capacity" for diabetes care, characterized by the inadequate availability of essential medicines and limited access to diabetes testing services [50]. These systemic barriers underscore the urgent need for tailored, context-specific interventions to address both structural and operational deficiencies in diabetes care in Ghana and other sub-Saharan Africa countries.

Our narrative synthesis revealed significant associations between suboptimal glycemic control and female gender, older age, higher income, and Akan ethnicity. Our finding on older age and the reduced risk for suboptimal glycemic control is supported by previous studies both within and outside the African continent [51–53]. For instance, a recent cross-sectional study among adults with T2D in Guinea and Cameroon (N = 1267) found that participants who were aged <65 years had 40% greater odds of having suboptimal glycemic control compared with those who were at least 65 years old [53]. In contrast, a few other studies have reported greater risk for suboptimal glycemic control with increasing age [9]. These inconsistencies may be attributed to variations in how suboptimal glycemic control is defined across studies

– some using HbA1c levels, while others rely on fasting blood sugar (FBS) measurements. Additional studies are required to provide definitive and robust evidence on the topic. Although higher income has been shown in previous studies to be associated with better glycemia [9,54,55], we noted contrary findings in the current review. A possible explanation may be that individuals with higher income in Ghana are more likely to adopt urbanized lifestyles—characterized by increased consumption of energy-dense foods, reduced physical activity, and greater levels of stress—which may contribute to poorer glycemic outcomes [56].

Regarding gender, whereas our narrative synthesis found that women were less likely to have suboptimal glycemic control, our subgroup meta-analysis revealed no significant differences in the prevalence of suboptimal glycemic control between men and women. These findings do not align with previous meta-analyses, which report a significantly higher risk of suboptimal glycemic control in women [6,51]. This lack of consistency may be attributed to the limited number of studies on sociodemographic determinants, which constrained our ability to robustly pool the data. Gender differences in the prevalence of poor glycemic control may be partly explained by differences in behavioral outlooks toward medicine. For example, in Ghana, previous studies have shown that men are generally more likely to engage in self-monitoring of blood glucose compared to women [13,57]. These behavioral differences should be considered when designing tailored interventions to improve glycemic control in this population.

While the HbA1c and FBG cut-off values for suboptimal glycemic control are based on the World Health Organization's definitions, several studies have questioned the applicability of these thresholds for individuals of African descent [58–60]. For example, a scoping review assessing the performance of HbA1c among individuals of African ancestry found that the standard diagnostic thresholds may lead to underdiagnosis of diabetes in African populations [58]. Despite this, the Ghanaian Ministry of Health continues to rely on these standard cut-offs, even though robust, context-specific validation studies are lacking [61]. We recommend that future studies examine whether these diagnostic thresholds are strongly associated with cardiometabolic risk profiles in Ghanaian and other African populations. Such evidence is critical to inform potential revisions to clinical guidelines and improve early detection and management of diabetes in these settings.

The findings of this study have important implications for public health policy and care delivery in Ghana. The high prevalence of suboptimal glycemic control across regions highlights the urgent need for nationwide interventions to address systemic barriers to effective diabetes management. Key areas for improvement include increasing access to essential medications, expanding diabetes testing and monitoring services, and enhancing healthcare infrastructure. Furthermore, policy reforms should prioritize building the capacity of healthcare providers through targeted training programs and professional development initiatives, ensuring that healthcare workers are equipped with the necessary skills and knowledge to deliver optimal diabetes care. Lastly, future studies should consider validating standard HbA1c and FBG cut-off values or developing new context-specific diagnostic thresholds for use in the Ghanaian population.

### Strengths and limitations of study

This study presents the first meta-analysis to estimate the prevalence of suboptimal glycemic control among individuals with diabetes in Ghana, providing valuable insights into the country's diabetes management challenges. The robustness of the analysis was ensured through sensitivity checks and the trim-and-fill method, which also addressed potential publication bias. By including both FBG and HbA1c metrics, the study accommodates variations in diagnostic practices, making the results relevant to diverse clinical and resource settings. Furthermore, subgroup analyses provided additional insights into demographic variations in glycemic control, contributing to a more nuanced understanding of the issue.

However, the study is not without its limitations. First, certain regions, such as the Northern Belt, were underrepresented in the studies included in the meta-analysis, which may limit the generalizability of the findings. Second, the scarcity of studies examining sociodemographic determinants restricted the ability to conduct meta-analyses on the factors

associated with suboptimal glycemic control. Third, while our electronic search was comprehensive, the exclusion of grey literature from this review might have resulted in the omission of potentially valuable data points particularly given the limited number of peer-reviewed studies on this topic in Ghana. Lastly, while adjustments for publication bias were made, other residual biases, including those arising from data reporting, could still influence the results.

## Conclusions

This study reveals a high prevalence of suboptimal glycemic control among individuals with diabetes in Ghana, with an overall rate of 67.6%. This aligns with regional trends across sub-Saharan Africa, underscoring shared systemic challenges in diabetes care. Sociodemographic factors such as age, income, gender, and ethnicity were found to be associated with suboptimal glycemic control. However, these associations were drawn from a limited number of studies, highlighting the need for further research to strengthen the evidence base and inform the development of effective, context-specific interventions.

In the interim, immediate action is needed to improve diabetes management by increasing access to affordable care, strengthening healthcare systems, and investing in training programs to enhance the capacity of healthcare providers. These efforts, combined with evidence-based, locally tailored interventions, can help mitigate the high burden of suboptimal glycemic control and its associated complications, ultimately improving health outcomes for individuals with diabetes in Ghana.

## Supporting information

**S1 Checklist. PRISMA 2020 checklist.**
(DOCX)

**S1 Text. Database search strategy.**
(DOCX)

**S1 Table. Quality assessment of included studies.**
(DOCX)

**S1 Fig. Subgroup analysis of prevalence estimates.**
(DOCX)

**S2 Fig. Forest plot for trim-and-fill analysis.**
(DOCX)

## Author contributions

**Conceptualization:** Emmanuel Ekpor, Dorothy Wilson, Eric Peprah Osei, Bernard Abeiku Mensah.

**Data curation:** Emmanuel Ekpor, Eric Peprah Osei, Bernard Abeiku Mensah.

**Formal analysis:** Emmanuel Ekpor, Dorothy Wilson, Eric Peprah Osei, Bernard Abeiku Mensah.

**Investigation:** Emmanuel Ekpor, Eric Peprah Osei.

**Methodology:** Emmanuel Ekpor, Dorothy Wilson, Bernard Abeiku Mensah, Samuel Akyirem.

**Resources:** Emmanuel Ekpor.

**Writing – original draft:** Emmanuel Ekpor, Eric Peprah Osei, Bernard Abeiku Mensah.

**Writing – review & editing:** Emmanuel Ekpor, Dorothy Wilson, Eric Peprah Osei, Bernard Abeiku Mensah, Samuel Akyirem.

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
