## [Decision Letter · Decision Letter 0]

Dear Dr. Akyirem,

Thank you for submitting your manuscript to PLOS ONE. After careful consideration, we feel that it has merit but does not fully meet PLOS ONE’s publication criteria as it currently stands. Therefore, we invite you to submit a revised version of the manuscript that addresses the points raised during the review process.

We look forward to receiving your revised manuscript.

Kind regards,

Rajiv Janardhanan, Ph.D.

Academic Editor

PLOS ONE

2. As required by our policy on Data Availability, please ensure your manuscript or supplementary information includes the following:

Additional Editor Comments (if provided):

Reviewers' comments:

Reviewer's Responses to Questions

**Comments to the Author**

1. Is the manuscript technically sound, and do the data support the conclusions?

Reviewer #1: Yes

Reviewer #2: Yes

2. Has the statistical analysis been performed appropriately and rigorously?

Reviewer #1: Yes

Reviewer #2: Yes

3. Have the authors made all data underlying the findings in their manuscript fully available?

Reviewer #1: Yes

Reviewer #2: No

4. Is the manuscript presented in an intelligible fashion and written in standard English?

Reviewer #1: Yes

Reviewer #2: Yes

Reviewer #1: The manuscript provides a valuable contribution to diabetes research in Ghana and Sub-Saharan Africa, given the increasing prevalence of diabetes in the region. The study findings add to the existing literature on the prevalence and sociodemographic determinants of suboptimal glycemic control among individuals with diabetes in Ghana. The authors have followed the PRISMA guidelines and the methodology used for meta-analysis is sound. However, the manuscript requires minor revisions before it can be published.

Comment 1:

While the authors have done an exhaustive search of articles to be included in the study, why is it that they have excluded the gray literature? The authors have mentioned that there are limited studies conducted in Ghana on this topic, so an explanation as to why gray literature is excluded should be added in the methodology section.

Comment 2:

The authors have mentioned why they chose Ghana as the focus of their study by stating that there are limited studies on Ghana. However, they have also stated that Ghana mirrors other countries in Africa in terms of diabetic trends. Given this, they should clarify why they chose to focus solely on Ghana when they could have conducted a comparative analysis with other countries.

Comment 3:

The findings show that there are significant gender, income, and age differences in suboptimal glycemic control across studies. However, this is not emphasized in the discussion section. The authors can further elaborate on this and can cite other studies that have tried to examine this relationship. A detailed paragraph on why these inconsistencies exist can help better understand the socio-demographic determinants of suboptimal glycemic control.

Comment 4:

While the authors have focussed specifically on Ghana, they have not suggested the measures to tackle this problem. The authors need to elaborate why Ghana is different from other countries and propose strategies to address this issue.

Reviewer #2: The report by Ekpor et. al. contains meta-analysis of diabetes management landscape for Ghana.

Figure 1: Authors should mention reason for exclusion of 588 records and mention this information in text as well.

“whiles > 7.0/ ≥7.0% (n=10) was used for HbA1c” should be “whiles> 7.0/ ≥7.0% (n=10) was used for HbA1c”

In Table 1, the column for Diabetes type and Design can be removed. The information under this column can be provided as footnotes with appropriate symbols so that the table appears within the margins. Similarly, the Author (year) can be changed to “Ref” and a numerical reference added.

It is not clear how the points were assigned for “quality assessment”. Authors should describe the quality assessment method. Instead of referring readers to supplementary information. Supp Table 1 should be referenced in Methods section as well.

Authors state that only 4 studies investigated sociodemographic determinants, Figure S1 have more than 4 studies listed for the gender analysis. Pl explain the statement or rephrase it.

There is a lack of discussion on the behavioral outlook towards medicine in both genders.

Authors are taking the glucose levels that are designed and created for a different race and continent.

Given authors have repeatedly mentioned “limited capacity”. There is a lack of discussion on what should be Ghana-specific glucose levels beyond what the physicians are calling diabetes. Diabetes and response to diabetes is heavily driven by cultural practices, authors fail to discuss this.

Along with generic recommendations provided by authors, they should recommend to developing Ghana-specific diagnostic glucose levels, which they discussed as a limitation.

The data availability statement should be modified to state that data collected is available upon request to the corresponding author.

**Do you want your identity to be public for this peer review?** For information about this choice, including consent withdrawal, please see our Privacy Policy

Reviewer #1: No

Reviewer #2: **Yes: ** Sharad Purohit

---

## [Author Response · Author response to Decision Letter 1]

18 Apr 2025

Dear Editor,

Thank you for the opportunity to revise our manuscript entitled “Prevalence and sociodemographic determinants of suboptimal glycemic control in persons with diabetes in Ghana: a systematic review and meta-analysis,” which we submitted to the PLOS One. We appreciate the time and effort that the editor and reviewers have taken to evaluate our work and provide constructive feedback.

We have carefully considered the insightful comments made by the reviewers and have made corresponding amendments to reflect the suggestions made. Changes to the manuscript body are highlighted in yellow and our responses to the reviewers’ comments summarized below.

Reviewer #1

Comment 1: While the authors have done an exhaustive search of articles to be included in the study, why is it that they have excluded the gray literature? The authors have mentioned that there are limited studies conducted in Ghana on this topic, so an explanation as to why gray literature is excluded should be added in the methodology section.

Authors’ response: We thank the reviewer for noting this. We agree that gray literature may be an important source of additional data, however, we chose to exclude it from this review to maintain a consistent standard of peer-reviewed quality across all included studies. Additionally, articles from gray literature (e.g. posters) often lack adequate detail on study design, which could limit comparability. Nonetheless, we recognize this as a limitation and have now noted it as such in the discussion section of the manuscript.

Comment 2: The authors have mentioned why they chose Ghana as the focus of their study by stating that there are limited studies on Ghana. However, they have also stated that Ghana mirrors other countries in Africa in terms of diabetic trends. Given this, they should clarify why they chose to focus solely on Ghana when they could have conducted a comparative analysis with other countries.

Authors’ response: We thank the reviewer for this comment. We chose to focus solely on Ghana because our review was intended to answer a focused research question about glycemic control in Ghana. By focusing on Ghana alone, we were able to conduct granular analysis highlighting within-country regional differences in glycemic control. We understand that including other African countries would have allowed for comparative analysis. However, these comparisons have been highlighted in the discussion section where we compared our results with findings from Ethiopia and other sub-Saharan African countries.

Comment 3: The findings show that there are significant gender, income, and age differences in suboptimal glycemic control across studies. However, this is not emphasized in the discussion section. The authors can further elaborate on this and can cite other studies that have tried to examine this relationship. A detailed paragraph on why these inconsistencies exist can help better understand the socio-demographic determinants of suboptimal glycemic control.

Authors’ response: We thank the reviewer for their feedback. We have added two paragraphs to the discussion to elaborate on these findings.

Comment 4: While the authors have focused specifically on Ghana, they have not suggested the measures to tackle this problem. The authors need to elaborate why Ghana is different from other countries and propose strategies to address this issue.

Authors’ response: We have now highlighted some strategies to address the issue of suboptimal glycemic control in the discussion section.

Reviewer #2

Comment 1: Figure 1: Authors should mention reason for exclusion of 588 records and mention this information in text as well.

Response: Thank you for the comment. We have now amended the figure and text to reflect this suggestion.

Comment 2: “whiles > 7.0/ ≥7.0% (n=10) was used for HbA1c” should be “whiles> 7.0/ ≥7.0% (n=10) was used for HbA1c”

Authors’ response: We amended the text to reflect this suggestion.

Comment 3: In Table 1, the column for Diabetes type and Design can be removed. The information under this column can be provided as footnotes with appropriate symbols so that the table appears within the margins. Similarly, the Author (year) can be changed to “Ref” and a numerical reference added.

Authors’ response: We thank the reviewer for the thoughtful comments. We believe that adding symbols for diabetes type and study design may make the table overly complicated and difficult to read. We agree with the reviewer’s concerns about the tables appearing in the margins. We have now addressed this by placing the table in a landscape layout.

Comment 4: It is not clear how the points were assigned for “quality assessment”. Authors should describe the quality assessment method. Instead of referring readers to supplementary information. Supp Table 1 should be referenced in Methods section as well.

Authors’ response: We have clarified in the methods that we determined the total quality score by counting the number of “yes” responses. The results of the quality assessment are explained on page 11 of the manuscript. We have also now referenced Supp Table 1 in the methods section as well.

Comment 5: Authors state that only 4 studies investigated sociodemographic determinants, Figure S1 have more than 4 studies listed for gender analysis. Pl explain the statement or rephrase it.

Authors’ response: We thank the reviewer for their comment. The four studies included in the narrative review were the ones that presented data on the association between glycemic control and gender using odds ratios. We have modified the sentences to enhance clarity. The studies in Figure S1 are those that stratified the prevalence of suboptimal glycemic control by gender, allowing us to calculate the prevalence for male vs females.

Comment 6: There is a lack of discussion on the behavioral outlook towards medicine in both genders.

Authors’ response: We have now included a brief discussion of the behavioral outcome in both genders.

Comment 7: Authors are taking the glucose levels that are designed and created for a different race and continent.

Authors’ response: We agree with this point. However, given that this is a systematic review, we are limited to the blood glucose categories or levels currently used in the published literature. The values we chose are also in line with the diabetes treatment guidelines from Ghana’s Ministry of Health. We have also mentioned this in the discussion section.

Comment 8: Given authors have repeatedly mentioned “limited capacity”. There is a lack of discussion on what should be Ghana-specific glucose levels beyond what the physicians are calling diabetes. Diabetes and response to diabetes is heavily driven by cultural practices, authors fail to discuss this.

Authors’ response: We do not make specific recommendations for Ghana-specific glucose levels as that is beyond the scope of this study. However, we have emphasized the need for further studies to validate these cut-off values for blood glucose in Ghanaian and African settings.

Comment 9: Along with generic recommendations provided by authors, they should recommend to developing Ghana-specific diagnostic glucose levels, which they discussed as a limitation.

Authors’ response: We have added the need to develop Ghana-specific glucose levels to the discussion.

Comment 10: The data availability statement should be modified to state that data collected is available upon request to the corresponding author.

Authors’ response: We have amended the statement to reflect the reviewer’s suggestion.

---

## [Decision Letter · Decision Letter 1]

Prevalence and sociodemographic determinants of suboptimal glycemic control in persons with diabetes in Ghana: a systematic review and meta-analysis

PONE-D-25-02107R1

Dear Dr. Akyirem,

We’re pleased to inform you that your manuscript has been judged scientifically suitable for publication and will be formally accepted for publication once it meets all outstanding technical requirements.

Kind regards,

Rajiv Janardhanan, Ph.D.

Academic Editor

PLOS ONE

Additional Editor Comments (optional):

Reviewers' comments:

Reviewer's Responses to Questions

**Comments to the Author**

Reviewer #2: All comments have been addressed

2. Is the manuscript technically sound, and do the data support the conclusions?

Reviewer #2: Yes

3. Has the statistical analysis been performed appropriately and rigorously?

Reviewer #2: Yes

4. Have the authors made all data underlying the findings in their manuscript fully available?

Reviewer #2: No

5. Is the manuscript presented in an intelligible fashion and written in standard English?

Reviewer #2: Yes

Reviewer #2: (No Response)

**Do you want your identity to be public for this peer review?** For information about this choice, including consent withdrawal, please see our Privacy Policy

Reviewer #2: No

---

## [Editor Report · Acceptance letter]

PONE-D-25-02107R1

PLOS ONE

Dear Dr. Akyirem,

I'm pleased to inform you that your manuscript has been deemed suitable for publication in PLOS ONE. Congratulations! Your manuscript is now being handed over to our production team.

Kind regards,

on behalf of

Dr. Rajiv Janardhanan

Academic Editor

PLOS ONE